# A Concept Level Energy-Based Framework for Interpreting Black-Box Large Language Model Responses

## Abstract

The widespread adoption of proprietary Large Language Models (LLMs) accessed strictly through closed-access APIs has created a critical challenge for their reliable deployment: a fundamental lack of interpretability. In this work, we propose a model-agnostic, post-hoc interpretation framework to address this. Our approach defines an energy model that quantifies the conceptual consistency between prompts and the corresponding LLM-generated responses. We use this energy to guide the training of an interpreter network for a set of target sentences. Once trained, our interpreter operates as an efficient, standalone tool, providing sentence-level importance scores without requiring further queries to the original LLM API or energy model. These scores quantify how much each prompt sentence influences the generation of specific target sentences. A key advantage is that our framework globally trains a local interpreter, which helps mitigate common biases in LLMs. Our experiments demonstrate that the energy network accurately captures the target LLM's generation patterns. Furthermore, we show that our interpreter effectively identifies the most influential prompt sentences for any given output.

## 1 Introduction

In recent years, the extraordinary performance of Large Language Models in complex tasks has encouraged machine learning researchers and developers to adopt them in different applications. Powerful LLMs are mostly provided as APIs by developer companies, and the detailed architectures and pre-training datasets are often unavailable. Additionally, when the architecture and dataset are available, their complexity prevents an exact understanding of how outputs are generated. In high-stakes domains such as medicine and law, this opacity prevents human experts from verifying a model's reasoning against domain knowledge and discovering hidden biases, thereby hindering its ability to satisfy the application-grounded evaluation criteria necessary for responsible deployment (Doshi-Velez & Kim, 2017).

Post-hoc attribution interpretation attempts to explain the behavior of trained machine learning models by finding an importance vector for input features. These methods measure how much each input feature affects the value of the output in each locality of the input space. White-box techniques that rely on gradients or internal activations are immediately disqualified for most real-world scenarios due to the lack of model access. Moreover, the faithfulness of popular proxies like attention weights has been rigorously challenged (Jain & Wallace, 2019). Various techniques have also been introduced for attribution-based interpretation of black-box models (Ribeiro et al., 2016; Lundberg & Lee, 2017; Seyyedsalehi et al., 2022). Most of these methods are developed to explain discriminative models with well-defined vectorized outputs. The problem of interpretation for generative models is fundamentally ill-posed. Generative models learn complex and implicit representations to produce high-dimensional and multifaceted outputs like text. Therefore, explaining them requires grappling with the complexity of their interactive outputs and the sheer volume of information in each generation (Schneider, 2024)

The exact goal of an interpretation method for an LLM is not well-defined and depends on the application. However, an attribution method to interpret an LLM is expected to relate elements of

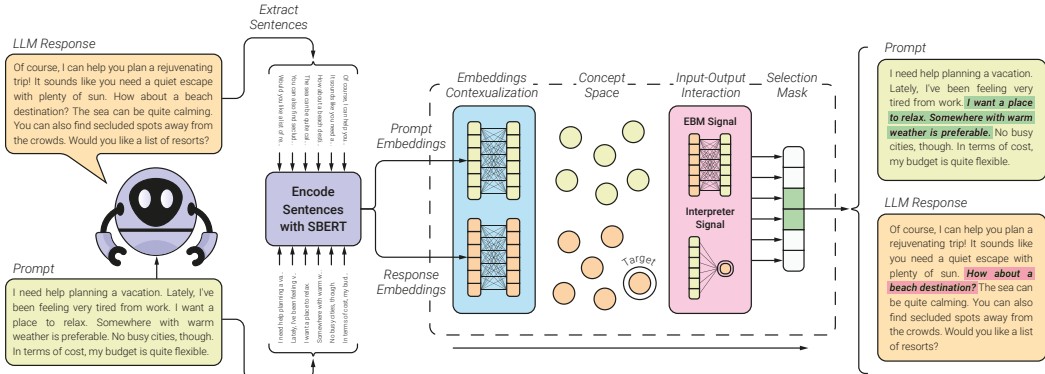

Figure 1: The prompt and response of the target black-box LLM are split in a sentence-wise manner and given to a pre-trained sentence embedding module. Embeddings are contextualized and transferred to a concept space, which simulates the thought process of the LLM. This concept space is trained using signals from an energy network. The proximity of sentences in the concept space demonstrates the influence of prompt sentences in generating output sentences.

the output text to the user prompt. Here, we consider both input and output texts at a sentence-level resolution. As the target output, we focus on a subset of output sentences and attempt to find an importance vector for the prompt sentences which shows how much each of the prompt sentences influenced the LLM to generate the target subset of the output. Then, we can find a subset of prompt sentences that were most influential in generating that target.

Prompt-based self-explanation, where an LLM is asked to justify its output using techniques like Chain-of-Thought (Wei et al., 2022), is prone to circular logic; it relies on the same, potentially biased, generative capabilities we seek to understand. Such explanations can produce plausible-sounding confabulations that are unfaithful to the model's true computational process, a form of motivated reasoning (Turpin et al., 2023). While efforts to steer model outputs via automated prompt engineering can overcome certain biases, these methods cast the problem as a search for an instruction that maximizes the probability of a known, pre-defined target answer (Zou et al., 2023; Zhou et al., 2023), which is almost unusable for interpretation tasks.

In this work, we propose a framework to train a post-hoc attribution interpreter for an arbitrary LLM. Here, we globally train a local interpreter. A local interpreter attempts to explain the behavior of the model on a per-instance basis. Local interpreters like LIME (Ribeiro et al., 2016) only observe samples from the neighborhood of the target instance during interpretation. However, this approach may not capture the global behavior of the complex model and can result in interpretability illusions (Friedman et al., 2024). By training globally, our interpreter observes various prompts and LLM responses, which helps it to capture the global behavior of the LLM and mitigate the effects of its intrinsic biases.

Fig. 1 illustrates an overview of the proposed approach. We train a transformer-based energy network over the sentences of a prompt and an LLM response. In this network, prompt and response sentences are mapped to a concept space that simulates input-output latent semantic relationships embedded in the target LLM. Then, the energy value is calculated based on the proximity of sentences in this concept space. The energy value can be considered as a metric that indicates how likely two texts are to be a prompt and its corresponding output from the target LLM. Using this metric, we train an interpreter for the target LLM, which takes the prompt and a subset of output sentences as input and returns an importance vector for the prompt sentences. This importance vector shows how much each prompt sentence influenced the LLM to generate the target subset of the output. Using this importance vector, the interpreter selects the subset of the most influential sentences in the prompt for the generation of the output. The key advantages of this work are as follows:

1. We shift the unit of analysis from noisy tokens to semantically coherent concepts, which we define as sentences, enabling a more human-intelligible level of attribution.

2. To operate in a black-box setting, we train a transformer-based energy-based model (EBM) that learns a random field over prompts and their corresponding LLM responses to simulate its thought process. This model successfully distinguishes responses of the target LLM from those written by humans or generated by other language models. As a surrogate for the target LLM, we use this EBM to guide the training procedure of an interpreter.

3. Finally, we propose a post-hoc, model-agnostic framework for interpreting black-box LLMs at a conceptual level. This interpreter finds the most influential sentences of the prompt, which triggered the LLM to generate the target subset of output sentences.

The remainder of this paper is organized as follows. Section 2 reviews related work, Section 3 details our proposed method, Section 4 presents our experimental results, and Section 5 concludes.

## 2 RELATED WORK

### 2.1 POST-HOC ATTRIBUTION METHODS TO EXPLAIN LANGUAGE MODELS

Input-output attribution methods aim to score the importance of input features for a given model output. A major line of work requires white-box access to model internals. Gradient-based methods compute saliency maps by propagating the output gradient back to the input tokens (Simonyan et al., 2014; Sundararajan et al., 2017; Shrikumar et al., 2017; Chefer et al., 2021). Attention-based methods propose using the model's internal attention weights as a direct proxy for feature importance (Xu et al., 2015; Li et al., 2017; Xie et al., 2017; Hao et al., 2021). However, this approach has been criticized for its lack of faithfulness, as attention scores do not always correlate with feature importance measured by other means (Jain & Wallace, 2019).

To circumvent the need for model access, several paradigms have been developed. Perturbation-based methods offer a model-agnostic alternative by measuring the change in output when parts of the input are removed or altered (Ribeiro et al., 2016; Lundberg & Lee, 2017; Yin & Neubig, 2022). Inspired by this approach, one study finds the influence of individual words in a prompt given to an LLM to generate an output (Hackmann et al., 2024). However, making this approach scalable for generative tasks often incurs a high computational cost, requiring thousands of model queries for a single explanation (Enouen et al., 2024; Zhao & Shan, 2024).

Finally, prompt-based self-explanation uses the LLM's own generative capabilities to produce a rationale, most notably through Chain-of-Thought (CoT) prompting (Wei et al., 2022). This approach is compelling but lacks guarantees of faithfulness, as the generated explanation may not reflect the model's true internal computation path but rather a plausible post-hoc rationalization (Turpin et al., 2023).

### 2.2 ENERGY-BASED MODELS IN NATURAL LANGUAGE PROCESSING

Energy-Based Models (EBMs) have been successfully adapted for generative NLP, primarily through using their ability to learn a global, sequence-level scoring function. One study demonstrated that a Transformer-based discriminator, trained to distinguish between human and machine-generated text, can effectively function as an EBM that assigns low energy to natural, coherent sequences (Bakhtin et al., 2019). This established the potential of EBMs as powerful, holistic text evaluators. One paradigm uses EBMs to refine the output of existing models. The Residual EBM approach adds a corrective energy term to the log-probabilities of a base autoregressive model, allowing the EBM to focus on capturing high-level properties like coherence that the base model may lack (Deng et al., 2020; Bakhtin et al., 2021). In another approach, the EBM is used as a post-processing reranker, which scores results of a base model to select the highest-quality one (Bhattacharyya et al., 2021). The work by (Tu et al., 2020) uses a powerful autoregressive model as a teacher to define an energy landscape; a student network is then trained via knowledge distillation to directly generate outputs that minimize this energy.

### 2.3 CONCEPT-BASED EXPLANATIONS

Growing consensus in interpretability research suggests that token-level attributions are often too granular for human understanding. This has prompted a shift toward concept-based explanations

that map model decisions to higher-level, human-intelligible ideas rather than individual features (Kim et al., 2018). Our work aligns with this paradigm by defining the sentence as the fundamental conceptual unit, as it represents a complete, robust thought for interpretation.

Treating sentences as coherent semantic units is well-justified by the evolution of language models. Foundational architectures like BERT were pre-trained with a Next Sentence Prediction (NSP) task to understand logical sentence relationships (Devlin et al., 2019). Subsequent work, such as Sentence-BERT, confirmed that fine-tuned sentence-level representations map similar meanings to nearby points in a vector space, establishing sentences as distinct semantic objects (Reimers & Gurevych, 2019). By leveraging sentences as our conceptual unit, our work is situated recent architectural innovations, such as Large Concept Models (LCMs), which propose shifting the core computational unit from tokens to sentence-level representations (Barrault et al., 2024).

## 3 METHODOLOGY

Our goal is to develop a post-hoc, model-agnostic method to interpret the response of any black-box language model to a specific prompt. When analyzing LLMs, working at the level of tokens is often suboptimal and computationally challenging. While individual words or tokens can be ambiguous, a sentence is the smallest unit of language that expresses a complete thought, proposition, or idea. By treating each sentence as a fundamental concept, we can analyze the model's reasoning at a more abstract and human-understandable level, focusing on the interplay of complete ideas rather than fragmented tokens.

Let $\mathbf{x}$ be a prompt to the LLM and $\mathbf{y}$ be the corresponding output. Also, let $\mathbf{y}_T$ denote the target subset of output sentences for which the interpreter should quantify the influence of each prompt sentence. To this end, we propose a two-stage framework: First, we pre-train an Energy-Based Model (EBM), $\mathcal{E}_{\mathrm{LM}}(\mathbf{x}, \mathbf{y}; \theta)$, that evaluates the consistency of a pair $(\mathbf{x}, \mathbf{y})$ with the generation pattern of the target LLM. The energy model is a function with the set of parameters $\beta$ which outputs a scalar value. Second, we use this EBM to guide the training of a lightweight interpreter, $\mathcal{IN}(\mathbf{x}, \mathbf{y}_T; \alpha)$, with the set of parameters $\alpha$. The interpreter is a function that takes the prompt, the LLM response, and the indices of the target sentences in the output, and returns a binary vector whose size equals the number of prompt sentences. In this vector, the values of 1 indicate selected sentences of the prompt as the most influential in generating the target.

### 3.1 PRE-PROCESSING WITH SENTENCE-BERT

The first step in our pipeline is to pre-process both the input text $\mathbf{x}$ and the output text $\mathbf{y}$ by segmenting them into sequences of sentences. We employ a pre-trained sentence transformer, such as Sentence-BERT (Reimers & Gurevych, 2019), as a frozen embedding module. This module maps each sentence to a fixed-dimensional embedding, producing initial representations. Finally, the set of embeddings for the prompt sentences, $S^{\mathrm{in}}$, and output sentences, $S^{\mathrm{out}}$, are passed to the energy and interpreter models.

### 3.2 THE ENERGY-BASED SURROGATE MODEL

To approximate the behavior of the black-box LLM, we design a globally-aware EBM, $\mathcal{E}_{\mathrm{LM}}(\mathbf{x}, \mathbf{y}; \theta)$, that learns to distinguish the most likely target-LLM-generated pairs from inauthentic ones using an energy score. The lower the assigned energy, the more likely the pair is authentic and consistent with what the target LLM would generate. This EBM serves as a differentiable and lightweight surrogate that captures the underlying logic of the LLM.

Figure 2a shows the architecture of the proposed EBM. In this module, sentence embeddings are processed as follows:

1. **Concept Space Projection:** Embeddings of input and output sentences, $S^{\mathrm{in}}$ and $S^{\mathrm{out}}$, are passed through separate, trainable self-attention modules ($\mathcal{P}^{\mathcal{E}}_{\mathrm{in\text{-}concept}}$ and $\mathcal{P}^{\mathcal{E}}_{\mathrm{out\text{-}concept}}$). These modules contextualize each sentence embedding with respect to its surrounding sentences, projecting them into what we term a concept space, resulting in $C^{\mathrm{in}}$ and $C^{\mathrm{out}}$. The concept space captures the LLM's internal reasoning process. Within this space, the distance

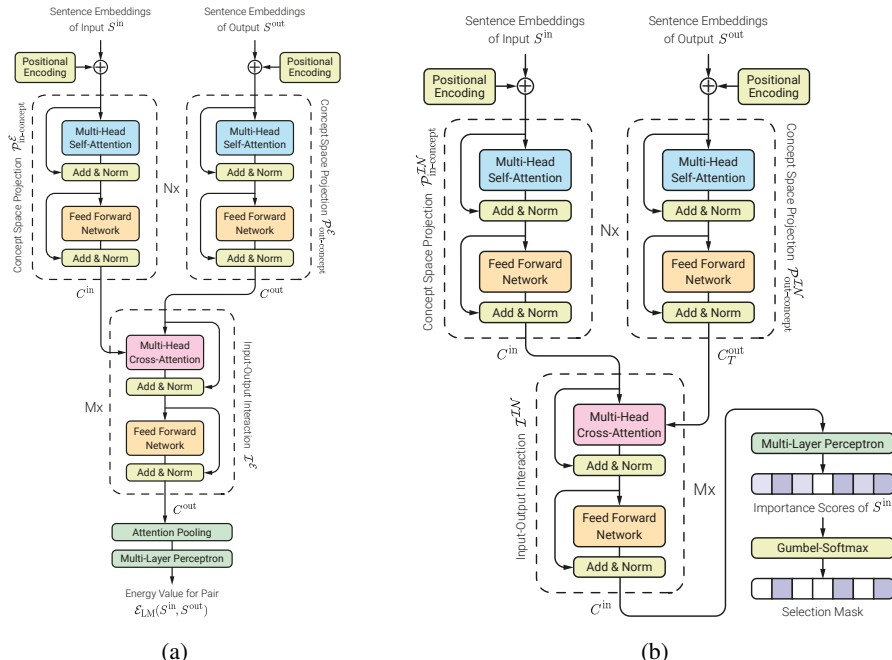

Figure 2: Proposed architectures of (**a**) the energy network and (**b**) the interpreter network

between sentences is determined by the LLM's inherent cognitive patterns, which are a function of its underlying architecture and training dataset.

2. **Input-Output Interaction:** To model the influence of the input on the output, the output concepts $C^{\text{out}}$ attend to the input concepts $C^{\text{in}}$ through a cross-attention block. This allows the model to weigh the relevance of each input concept to the overall output.

3. **Energy Calculation:** The resulting representations are aggregated via attention pooling and passed through a final multi-layer perceptron (MLP), which outputs a single scalar energy value $\mathcal{E}_{\text{LM}}(\mathbf{x}, \mathbf{y}; \theta)$.

The EBM network is trained in two phases. First, it is pre-trained using an expanded InfoNCE loss. Then, it is fine-tuned during the training of the interpreter. In the pre-training phase, we generate a dataset of a set of prompts and their corresponding outputs $(\mathbf{x}, \mathbf{y})$, by querying the target black-box LLM.

To apply the InfoNCE loss, we generate a rich set of negative samples for each positive one. In a **structural corruption** setup, For a pair $(\mathbf{x}, \mathbf{y})$, we create two types of negatives: $(\mathbf{x}, \mathbf{y}')$ and $(\mathbf{x}', \mathbf{y})$. The corruptions $\mathbf{x}'$ and $\mathbf{y}'$ are formed by either masking entire sentences (to teach concept continuity) or masking individual tokens (to teach grammatical structure). These negative samples encourage the energy function to assign higher energy to less reasonable sentences. In a **semantic-based** negative sampling, to learn the style and context of pairs, we also construct negatives using outputs from other models or humans, as well as through *off-topic sampling* to switch correct pairs. In the latter, for a positive pair $(\mathbf{x}_i, \mathbf{y}_i)$, we generate negative samples of the form $(\mathbf{x}_i, \mathbf{y}_j)$ and $(\mathbf{x}_j, \mathbf{y}_i)$ for $j \neq i$. This encourages the energy function to capture conceptual inconsistencies and approximate the target LLM's generation distribution by forcing it to distinguish authentic outputs from alternatives, thereby sharpening its discrimination boundary for better guidance of the interpreter.

Finally, the energy model is pre-trained to assign a lower energy score to a positive pair compared to all negative ones by the following loss function:

$$\mathcal{L}_{\mathcal{E}}^{PT} = -\log\left(\frac{\exp(-\mathcal{E}_{\text{LM}}(\mathbf{x}_i, \mathbf{y}_i; \theta)/\tau)}{\exp(-\mathcal{E}_{\text{LM}}(\mathbf{x}_i, \mathbf{y}_i; \theta)/\tau) + \sum_{(\mathbf{x}', \mathbf{y}') \in \mathcal{N}_i} \exp(-\mathcal{E}_{\text{LM}}(\mathbf{x}', \mathbf{y}'; \theta)/\tau)}\right) \quad (1)$$

where $\mathcal{N}_i$ is the set of negative samples for the $i$-th pair and $\tau$ is a temperature hyperparameter. In appendix A, figure 5 describes the pre-training procedure of the energy block.

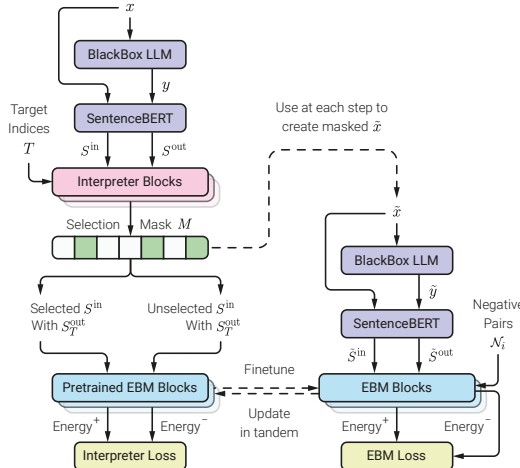

Figure 3: An overview of the EBM-guided training procedure for the interpreter. The framework involves a joint, alternating optimization where the interpreter's parameters are updated using a fixed EBM to create an input mask (left). Subsequently, the EBM is fine-tuned using the LLM's response to the masked prompt to mitigate distribution shift (right).

### 3.3 THE INTERPRETER MODEL

Given a prompt $\mathbf{x}$, the LLM response $\mathbf{y}$, and a subset of output sentences as target $\mathbf{y}_T$, the interpreter's objective is to quantify the influence of sentences in $\mathbf{x}$ in generating $\mathbf{y}_T$ and select the most important prompt sentences responsible for generating the target subset of the output.

Fig. 2b shows the architecture of the interpreter. First, embeddings of prompt sentences $S^{\text{in}}$ are projected to a concept space using a self-attention module $\mathcal{P}^{\mathcal{IN}}_{in-concept}$. Sentence embeddings of the output $S^{\text{out}}$ are also projected using another self-attention module $\mathcal{P}^{\mathcal{IN}}_{out-concept}$. Then, only the target sentences are retained and the others are masked. In the next step, the input concepts $C^{\text{in}}$ attend to target concepts $C^{\text{out}}_T$ via a cross-attention mechanism. The resulting attention weights are passed through an MLP layer, which results in a vector of importance scores. The last step of the interpreter module is a Gumbel-Softmax unit (Jang et al., 2016) by which we encourage the interpreter to find a specific number of the most important sentences for the target output. Details of this module are provided in appendix B.

Considering $\mathbf{x}$ as the prompt, we define,

$$\tilde{\mathbf{x}} = \mathbf{x} \odot \mathcal{IN}(\mathbf{x}; \mathbf{y}_T, \alpha) \tag{2}$$

where $\odot$ denotes the element-wise multiplication of the interpreter output with sentences of $\mathbf{x}$. As the energy function simulates the thought process of the LLM, if the interpreter successfully selects the most important sentences of the input for the target, the value of $\mathcal{E}_{\text{LM}}(\tilde{\mathbf{x}}, \mathbf{y}_T; \theta)$ should be at a minimum among the energy values for all other possible subsets of input sentences. Equivalently, as $\mathbf{x} - \tilde{\mathbf{x}}$ shows the least relevant parts of the prompt to the target, the value of $\mathcal{E}_{\text{LM}}(\mathbf{x} - \tilde{\mathbf{x}}, \mathbf{y}_T; \theta)$ should be at a maximum among the energy values for all other possible subsets of input sentences. If $\tilde{\mathbf{x}}$ were the most related subset of the prompt to the target output, the difference between these two values will be maximized. Therefore, we believe the ideal interpreter solves the following optimization problem:

$$\arg\max_{\alpha} \ \mathbb{E}_{(x,y)} \left[ \mathcal{E}_{\text{LM}}(\mathbf{x} - \tilde{\mathbf{x}}, \mathbf{y}_T; \theta) - \mathcal{E}_{\text{LM}}(\tilde{\mathbf{x}}, \mathbf{y}_T; \theta) \right] \tag{3}$$

#### 3.3.1 EBM-GUIDED TRAINING OF THE INTERPRETER

In equation 3, the energy value is calculated for the masked prompt and the LLM response. As mentioned in (Hsia et al., 2023), masking samples leads to a distribution shift compared to the original input distribution. Therefore, to mitigate the effect of the out-of-distribution problem, we fine-tune the energy network during the training of the interpreter with an InfoNCE loss.

Fig. 3 describes an overview of the proposed procedure to train the interpreter. The interpreter and EBM are optimized jointly in an alternating fashion, allowing them to adapt to each other and improve performance. The final optimization procedure is as follows:

1. **Update Interpreter:** In iteration $k$, for a fixed EBM $\mathcal{E}(.,.;\theta^{(k-1)})$, we update the interpreter's parameters to maximize equation 3 as follows:

$$\alpha^{(k)} \leftarrow \alpha^{(k-1)} + \beta \, \nabla_\alpha \mathbb{E}_{(x,y)} [\, \mathcal{E}_{\text{LM}} \left( \mathbf{x} - \mathbf{x} \odot \mathcal{IN}(\mathbf{x}; \mathbf{y}_T, \alpha), \, \mathbf{y}_T; \, \theta^{(k-1)} \right) -$$
$$\mathcal{E}_{\text{LM}} \left( \mathbf{x} \odot \mathcal{IN}(\mathbf{x}; \mathbf{y}_T, \alpha), \, \mathbf{y}_T; \, \theta^{(k-1)} \right) \,] \quad (4)$$

2. **Getting LLM response to the selected subset of the prompt:** The prompt is masked with the output of the current interpreter. Then, the response of the LLM is obtained for the masked prompt:

$$\tilde{\mathbf{x}} = \mathbf{x} \odot \mathcal{IN}(\mathbf{x}, \mathbf{y}_T; \alpha^{(k)}) \quad (5)$$
$$\tilde{\mathbf{y}} = \text{LM}(\tilde{\mathbf{x}}) \quad (6)$$

3. **Update EBM:** Using the new sample $(\tilde{\mathbf{x}}, \tilde{\mathbf{y}})$, the energy function is fine-tuned to capture the LLM behaviour in unexplored areas and avoid the distribution-shift problem:

$$\theta^{(k)} \leftarrow \theta^{(k-1)} +$$
$$\beta' \nabla_\theta \log \left( \frac{\exp(-\mathcal{E}_{\text{LM}}(\tilde{\mathbf{x}}, \tilde{\mathbf{y}}; \theta)/\tau)}{\exp(-\mathcal{E}_{\text{LM}}(\tilde{\mathbf{x}}, \tilde{\mathbf{y}}; \theta)/\tau) + \sum_{(\mathbf{x}', \mathbf{y}') \in \mathcal{N}} \exp(-\mathcal{E}_{\text{LM}}(\mathbf{x}', \mathbf{y}'; \theta)/\tau)} \right)$$
$$(7)$$

where **LM** denotes the target LLM, $\beta$ and $\beta'$ are learning rates and $\tau$ is the InfoNCE loss temperature. $\mathcal{N}$ shows the set of negative samples generated for the pair $(\tilde{\mathbf{x}}, \tilde{\mathbf{y}})$. The energy network's parameters are initialized from the pre-trained model. In the interpreter network, parameters of the self-attention modules are initialized with those in the energy network and the cross-attention module is initialized randomly. During this procedure, the information from the energy concept space, which is extracted from the target LLM, is transferred to the interpreter concept space. After training, the proximity of sentences in the concept space of the interpreter depends on the LLM's thought process. Therefore, it acts as a standalone tool for interpreting the target LLM's prompts and responses.

## 4 Experiments

In this section, we present the empirical validation of our proposed framework. We first demonstrate the effectiveness of the EBM pre-training in creating a faithful surrogate model, and then proceed to evaluate the performance of the final interpreter.

### 4.1 Validating the EBM as a Faithful Surrogate

The first experiment validates that our EBM can be effectively pre-trained to serve as a surrogate for a target LLM. The goal is to show that the EBM learns a well-defined energy landscape, assigning low energy to authentic prompt-response pairs while assigning high energy to various forms of corrupted or mismatched pairs.

**Target LLM and Dataset.** Our objective is to model the energy surface of `GPT-4o mini`. We utilized Hello-SimpleAI/HC3 dataset, a corpus widely used for comparing human and LLM-generated text across muliple domains (e.g., science, finance, medicine, and open-domain QA). We used the "question" column from the training split to prompt our target LLM, `GPT-4o mini`, to generate positive samples. For negative sampling, we used the provided human answers and also generated responses from `GPT-2` for the same prompts. We trained two models of different sizes: a smaller EBM-167M on 12,000 samples and a larger EBM-181M on 20,000 samples. For each, 10% of the data was reserved for validation.

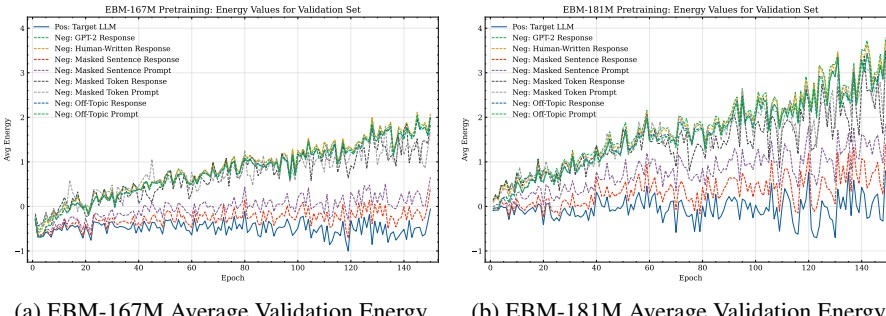

(a) EBM-167M Average Validation Energy  (b) EBM-181M Average Validation Energy

Figure 4: Averaged energy value for validation set for EBMs with two different sizes. The blue line shows the value for the target LLM response which successfully is the minimum one. For EBM-167M (**a**), negative samples become successfully distant from the positive one during training. For EBM-181M (**b**), we have a significantly wider energy gap, indicating enhanced discriminative power for larger network.

**Model Architecture.** We evaluated two EBM architectures, EBM-167M and EBM-181M, with 167 million and 181 million parameters, respectively. Both models use two concept space projection blocks for the input and output sentences, where each block consists of a self-attention layer with 8 heads and a feed-forward network. The key difference lies in the input-output interaction module; the EBM-167M uses four cross-attention blocks, while the more powerful EBM-181M uses six. Both architectures conclude with a two-layer MLP following the attention pooling stage.

**Pre-training and Negative Sampling.** The EBMs were pre-trained using the InfoNCE loss with a rich set of negative samples designed to teach different aspects of the target LLM's behavior. These included: **structural corruptions** (masking entire sentences or individual tokens in either the prompt or the response) and **semantic-based negatives** (swapping the response with one from a different model like `GPT-2`, a human-written answer, or an off-topic response from the model).

The EBM pre-training results, summarized in Figure 4 and corroborated by consistent loss convergence (Appendix A, Figure 6), show that both models successfully learned the target LLM's generation patterns. The validation energy plots reveal that the EBMs assign the lowest energy to authentic target samples, effectively distinguishing them from all negatives. As anticipated, semantic-based negatives (e.g., human or `GPT-2` responses) receive the highest energy, while structural corruptions like sentence masking proved to be the most challenging negative class, highlighting the model's sensitivity to nuanced, conceptual changes. Critically, distinguishing difficult off-topic negatives required careful tuning of model capacity and training data balance, with successful learning forcing the models beyond surface features to capture the essential semantic relationship between the prompt and response. Of the two, the larger EBM-181M trained on more data exhibited superior performance, showing a significantly wider energy gap (Figure 4b) than the EBM-167M. This superior, well-defined energy surface confirms the framework's scalability and potential for high-fidelity modeling of complex LLMs.

## 4.2 Interpretation Results

We train our interpreter model using guidance from the pre-trained energy block, EBM-181M (introduced earlier). The interpreter incorporates two concept-space projection blocks—one for input sentences and one for output sentences. Each projection block contains self-attention layers (initialized with the energy block weights to better align with concept spaces) and six cross-attention layers with eight attention heads, which is followed by a single MLP layer. To facilitate a more descriptive evaluation, we selected a movie review sentiment analysis task as the target LLM scenario. The goal is to identify which parts of each review most strongly influence the LLM's generated response. Specifically, we provide IMDB reviews (Lakshmi, 2020) to `GPT-4o mini` and ask it to produce a sentence summarizing the sentiment of the review. This generated sentiment sentence is then passed as the target output to the interpreter network. Table 1 presents some examples of post-hoc interpretation results produced by the interpreter model.

Table 1: Results of interpreting the `GPT-4o mini` responses to IMDB review prompts

| Review Full Text (Top Sentences Highlighted) | Predicted Sentiment |
|---|---|
| **Sample #1** | |
| This is an excellent film, with an extraordinary cast and acting. I was very disappointed when it didn't win the Academy Awards for best film and best actress (Whoopi Goldberg). It certainly deserved it. In any case, take a look at it; I am sure you will enjoy it very much. | Based on the given text, the review sentiment is **positive**. |
| **Sample #2** | |
| Some people might consider this movie a piece of artwork—to be able to express your imagination on film in order to create a movie filled with antagonizing pain and death. I personally think that this movie is a disgust, which should have never been released. This movie is repulsive, illogical, and meaningless. Not only is it a complete waste of time but it makes you sick for days to come. The appalling images shown in the film not only make you grasp for air but they set in your mind and it takes days to forget them. Such a shame that people waste their imagination on such inhumane suffering. "Kill Bill" would be another example, but at least "Kill Bill" has its purpose, meaning, climax, and resolution. | Based on the given text, the review sentiment is **negative**. |
| **Sample #3** | |
| Officially the first martial arts movie in USSR cinematography featuring actual martial artists like Tadeush Kas'yanov and Russian Bruce Lee - Talgat Nigmatullin. Bad people hijack a ship on the high seas, but fortunately, just about everybody on board is a trained martial artist. A collectible for martial arts aficionados. | Based on the given text, the review sentiment is **positive**. |
| **Sample #4** | |
| Slow, incomprehensible, boring. Three enthusiastic words that describe the movie of the book. This is surely a case where the movie should never have been made at the expense of the book. The best part of the movie was the scenery, excellent. The worst part was the slow moving interactions of the actors which combined with endless meaningful glances. The editing is abrupt and patchy. However, despite this, the actors worked very hard at least trying to be a little believable with a terrible script. It was startling that although set in Peru there was hardly a person of Peruvian descent wandering about the set—even in the flashback scenes depicting Peru in the 17th century. If you have any sense of history, try to avoid this movie. | Based on the given text, the review sentiment is **negative**. |

## 5  DISCUSSION AND CONCLUSION

The growing integration of generative models into critical applications heightens the demand for transparency and interpretability. In this work, we contribute to this effort by introducing a novel method for elucidating the reasoning of Large Language Models (LLMs) at a conceptual level. Results of our study provide important insights into the potential of energy-based models (EBMs) as a surrogate for black-box Large Language Models (LLMs) for further analysis. By employing a concept-level approach, we have demonstrated that the energy model successfully guides training of a model-agnostic, post-hoc input-output attribution method to interpret the LLM response at the sentence level.

Our method identifies the salient sentences in a prompt that trigger specific components of an LLM's response. This provides a critical lens for diagnosing model failures, such as hallucination and the activation of internal biases. Unlike local surrogate methods, our globally trained interpreter captures not only instance-specific relationships but also broader semantic patterns embedded in the LLM's generation process across the dataset. This offers a more profound insight into the model's decision-making mechanism.

We demonstrate that our approach is scalable, with larger EBMs (e.g., EBM-181M) yielding improved performance in distinguishing authentic from corrupted data pairs. A primary limitation is the computational cost associated with contrastive pre-training of the EBM, which can be prohibitive at scale. Future work will focus on mitigating this cost and extending the framework to multimodal settings, which will necessitate advances in cross-modal concept alignment.

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

APPENDIX

## A  ENERGY NETWORK PRE-TRAINING

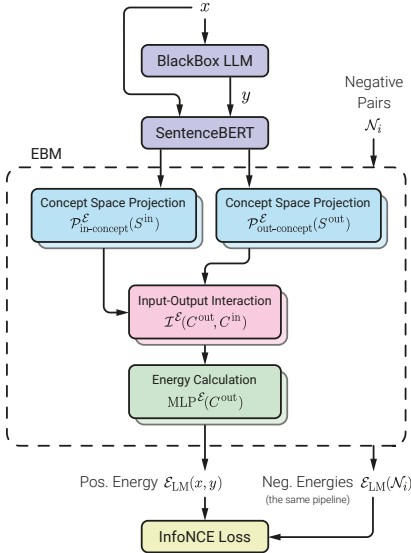

Figure 5: Pre-training pipeline of the globally-aware energy-based surrogate model, $\mathcal{E}_{\text{LM}}(\mathbf{x}, \mathbf{y}; \theta)$. Input and output sentences are first embedded via SentenceBERT and then projected into separate concept spaces through trainable self-attention modules. Output concepts cross-attend to input concepts to model input–output dependencies, after which an MLP computes a scalar energy score. Positive pairs $(\mathbf{x}, \mathbf{y})$ from the target LLM and negative pairs $\mathcal{N}_i$ are both scored, and the model is trained with an InfoNCE loss to assign low energy to authentic pairs and high energy to mismatched ones.

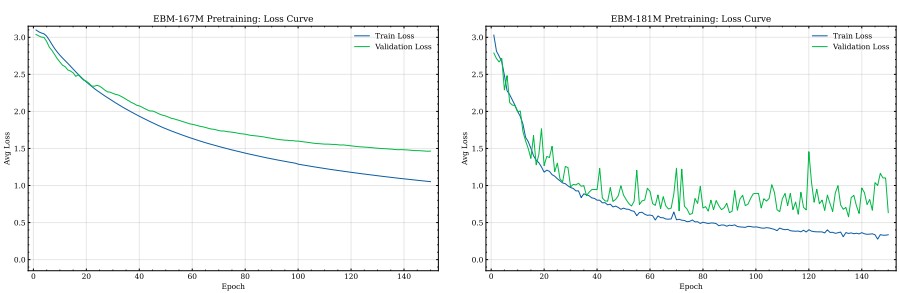

(a) EBM-167M Training and Validation Loss    (b) EBM-181M Training and Validation Loss

Figure 6: Pre-training Loss for both EBMs. (**a**) The EBM-167M training and validation loss shows steady convergence. (**b**) The larger EBM-181M loss decreases more rapidly, reflecting the benefits of increased capacity and data volume.

## B  DIFFERENTIABLE TOP-$K$ SENTENCE SELECTION VIA GUMBEL–SOFTMAX

The interpreter outputs a normalized importance scores with values between $0$ and $1$, that can be considered as a parameter of a Categorical distribution. However, we aim to select the top-$K$ most important ones by sampling from the interpreter's output, which is a non-differentiable operation. To address this, we employ the Gumbel-Softmax trick, which provides a continuous relaxation and thus enables differentiable training. In other words, we use a Gumbel-Softmax relaxation to highlight

the $K$ input sentences most related to a chosen subset of output sentences $Y$ in a trainable, end-to-end manner. Let an auxiliary scorer(Interpreter network in this case) produce relevance logits $z_i = \big(\mathcal{W}_\alpha(x, Y)\big)_i$ for the $n$ input sentences $x = (s_1, \ldots, s_n)$. We draw i.i.d. $u_i \sim \mathrm{Uniform}(0, 1)$ and form standard Gumbel noise

$$g_i = -\log\big(-\log u_i\big), \qquad i = 1, \ldots, n. \tag{A1}$$

With temperature $\tau > 0$, a relaxed one-hot $c \in \Delta^{n-1}$ is computed as

$$c_i = \frac{\exp\big((z_i + g_i)/\tau\big)}{\sum_{j=1}^n \exp\big((z_j + g_j)/\tau\big)}, \qquad i = 1, \ldots, n. \tag{A2}$$

As $\tau \to 0$, $c$ approaches a categorical sample from $\mathrm{softmax}(z)$; larger $\tau$ provides smoother assignments and stable gradients.

To obtain a $K$-hot selection over sentences, we draw $K$ independent relaxed samples $\{c^{(j)}\}_{j=1}^K$ using equation A1 & A2 and combine them elementwise by

$$m_i = \max_{j=1,\ldots,K} c_i^{(j)}, \qquad i = 1, \ldots, n, \tag{A3}$$

which serves as a continuous proxy for the top-$K$ indicator. Following the notation of the previous paper, the interpreter block output is

$$\mathcal{IN}_t(x; \alpha)_i = m_i = \max_{j=1,\ldots,K} c_i^{(j)}. \tag{A4}$$

During training, $\mathcal{IN}_t(x; \alpha)$ gates sentence representations while preserving gradients; at inference, we take the top-$K$ indices of $z$ (or harden the samples) to obtain discrete selections.

## C  THE LLM USAGE

Some parts of the initial drafts of this manuscript were revised with the assistance of a large language model. The model was prompted to improve the fluency, conciseness, and overall academic tone of the text to meet the standards of ICLR publications.

