# OpenReview forum: "A Concept Level Energy-Based Framework for Interpreting Black-Box Large Language Model Responses"
_ICLR.cc/2026/Conference — ICLR 2026 Conference Withdrawn Submission_

### Official Review · Reviewer_5awW · 2025-10-28

**Soundness:** 2
**Presentation:** 2
**Contribution:** 2
**Rating:** 2
**Confidence:** 5

**Summary:**

The paper proposes a framework for interpreting black-box large language models using a concept-level energy-based model (EBM) as a surrogate to approximate the model’s behavior. The EBM is trained on prompt–response pairs to learn an energy landscape that distinguishes authentic model outputs from mismatched or corrupted samples. An interpreter network is then trained, guided by the EBM, to identify which sentences in a prompt most influence a given output. Experiments include validating the EBM’s ability to differentiate valid from invalid pairs and a qualitative case study on sentiment analysis where the interpreter highlights key sentences associated with the model’s output sentiment.

**Strengths:**

1. The paper tries to tackle an important problem: interpreting black-box large language models at a concept level rather than through token-level attributions.

2. The proposed joint optimization between the EBM and the interpreter reflects an innovative attempt to train both models in a single framework.

**Weaknesses:**

While the paper proposes an interesting idea, the experimental section is not convincing to substantiate the claims.

1. The paper assumes that GPT-4o-mini responses are always superior and uses them as “positive” examples, while human-written or GPT-2 responses are treated as “negative.” This assumption is problematic since GPT-4o-mini outputs are not guaranteed to be more accurate or coherent, and human answers are not necessarily worse. The resulting supervision may capture stylistic or distributional differences rather than genuine semantic or factual correctness. The authors should justify this choice or adopt a quality-based labeling scheme instead of relying purely on model origin.

2. The proposed framework introduces multiple components such as the concept space projection, joint EBM-interpreter optimization, and contrastive pretraining, without providing theoretical motivation or ablation studies. It is unclear why the concept space projection is necessary on top of frozen Sentence-BERT embeddings, why joint optimization improves over independent training, or why InfoNCE contrastive loss is preferable to simpler alternatives. Without empirical evidence or rationale, these design choices appear ad hoc and unsubstantiated.

3. The experiments do not include comparisons with standard or simplified baselines, for example, a straightforward supervised transformer trained on the same data or existing post-hoc attribution methods (e.g., LIME, SHAP, or other EBM-based frameworks). Without such baselines, it is impossible to assess whether the proposed method provides any advantage in accuracy, interpretability, or efficiency.

4. The core claim that the interpreter effectively identifies influential prompt sentences is not empirically supported. The only quantitative result (Figure 4) measures energy separation during EBM pretraining, not the quality of interpretations. The IMDB sentiment case study is entirely qualitative and anecdotal, offering no metrics such as precision, recall, or correlation with human-annotated importance scores. Consequently, the interpretability claims remain unverified.

Overall, the paper’s experiments seem too limited to substantiate the strong claims made about interpretability and generalization. The presented results focus on verifying internal consistency rather than demonstrating practical usefulness or faithfulness. The work reads more like a proof-of-concept or early-stage project proposal than a fully validated research contribution. Substantial additional experiments like quantitative evaluations and ablations are needed to establish credibility.

**Questions:**

1. Could the authors provide a more formal definition of the “concept space”? How does it differ from a standard sentence embedding?

2. The paper lacks experiment and implementation details. For example, given the alternating updates between the EBM and the interpreter, how stable was the training process? Did you observe mode collapse, oscillations, or sensitivity to hyperparameters?

3. It would also strengthen the paper if the authors explicitly analyzed when and why the method fails.

---

### Official Review · Reviewer_9M1y · 2025-10-28

**Soundness:** 1
**Presentation:** 1
**Contribution:** 1
**Rating:** 2
**Confidence:** 4

**Summary:**

This paper proposes a two-stage, concept-level interpretability framework for black-box LLMs. First, an Energy-Based surrogate model is trained on sentence embeddings to assign low energy to authentic ⟨prompt, response⟩ pairs and high energy to negatives. Second, a lightweight interpreter guided by the EBM returns a binary mask of input sentences deemed most influential for a target output sentence.

**Strengths:**

It is a meaningful attempt to perform interpretability analysis on black-box models.

**Weaknesses:**

- The paper equates sentence-level influence selection with explaining LLM reasoning, but does not validate faithfulness or causal alignment to the LLM’s actual inference. The EBM demonstrates pairwise consistency discrimination not a mapping to the LLM’s reasoning steps.
- The only interpreter results are a few sentimental examples with highlighted sentences. There are **no quantitative metrics** and **no comparisons to any baselines**.
- While the architecture is described, the paper does not provide a testable definition linking the learned energy to reasoning processes or a decomposition connecting energy terms to human-interpretable concepts. The EBM is essentially an attention-pooled scoring network trained with InfoNCE.
- The interpreter is shown only on sentiment summarization, no evidence is given for multi-step reasoning, factuality, or chain-of-thought-like tasks.

**Questions:**

Authors are advised to give more experimental evidence to support this paper.

---

### Official Review · Reviewer_fzd1 · 2025-10-30

**Soundness:** 3
**Presentation:** 3
**Contribution:** 2
**Rating:** 6
**Confidence:** 4

**Summary:**

This paper proposes a framework for attributing a model’s output to specific input sentences within a prompt. By training a transformer-based energy network over the sentences of both input and output, the framework learns the semantic relationships between them in the embedding space. This enables the identification of the most influential subset of input sentences that contribute to the model’s response.

**Strengths:**

The paper extends attribution-based interpretability to the sentence level, which is a meaningful direction beyond token-level attribution. The proposed method is conceptually straightforward and well-motivated.

Experiments are conducted to demonstrate the interpreter model’s ability to identify meaningful input–output relationships, successfully distinguishing authentic from fabricated examples. The sentiment analysis case study further illustrates the interpreter’s capacity to attribute responses to interpretable input sentences. However, the case study remains relatively simple compared to real-world tasks such as chain-of-thought (CoT) reasoning.

Overall, this paper presents a novel and interesting approach for attributing model outputs to input sentences.

**Weaknesses:**

- The problem statement is not clearly articulated until Section 3, making it difficult for readers to grasp the main objective early on.
- The phrase “concept-level attribution” may be misleading—this work primarily addresses sentence-level attribution rather than conceptual abstraction.
- Influence function studies should be discussed as related work, as they are closely connected to input–output attribution and could help position this work more precisely within the interpretability literature.
- Lines 189–190: The term “indices of the target sentences in the output” is unclear. If this refers to parts of the input, it should be formally defined in the interpreter’s function.
- Lines 470–471: The claim about diagnosing model failures such as hallucination and internal bias seems overstated given the current experimental scope.

**Questions:**

- Section 3.1: When sentences are segmented using Sentence-BERT, how are their embeddings used for input and output? It appears they are treated sequentially as embeddings, but this should be clarified when first introduced.
- Section 3.3: Is the number of sentences in $\mathbf{x}$ fixed to allow elementwise multiplication? If so, please clarify this assumption.

---

### Official Review · Reviewer_KyCJ · 2025-10-31

**Soundness:** 2
**Presentation:** 2
**Contribution:** 2
**Rating:** 2
**Confidence:** 3

**Summary:**

This paper introduces a post-hoc, model-agnostic framework for interpreting LLMs. It trains a transformer-based energy model (EBM) to approximate the LLM’s reasoning process and identify which prompt sentences most influence specific output sentences. By learning sentence-level attributions across multiple prompts, the method provides global, concept-level explanations that capture the LLM’s overall behavior and biases, offering a more interpretable alternative to local or token-level attribution methods.

**Strengths:**

1. The paper introduces a creative, post-hoc, model-agnostic method for interpreting black-box LLMs, which does not require access to model internals.
2. The use of a transformer-based EBM as a differentiable surrogate to approximate LLM reasoning is methodologically interesting.

**Weaknesses:**

1. The framework depends on how accurately the EBM captures the target LLM’s internal logic. If the EBM fails to approximate the reasoning process well, the entire interpretation may be unreliable.
2. The two-stage training (EBM pretraining + interpreter fine-tuning) with alternating optimization increases computational cost and implementation complexity, limiting practical usability.
3. All experiments are conducted on GPT-4o mini and a sentiment analysis task. The lack of multi-domain or multi-model validation limits claims of generalizability and model-agnosticism.
4. The pre-training uses only 12K–20K samples from the HC3 dataset, which is small compared to standard LLM-scale interpretability settings. Additionally, only movie reviews are used for final interpretation results, lacking diversity in domains or prompt types.

**Questions:**

1. The paper claims that the Energy-Based Model (EBM) captures the generation dynamics of the target LLM. How do the authors ensure that the learned energy function truly reflects the reasoning or decision-making process of the black-box LLM rather than simply modeling surface-level text similarity?
2. The paper emphasizes the widening energy gap between positive and negative samples as evidence of better modeling. However, is there empirical evidence that a larger energy gap directly translates to more accurate or meaningful interpretation?
3. Since the framework involves both an EBM (up to 181M parameters) and an interpreter with attention layers, how does the computational cost compare with simpler post-hoc methods? Is it practical for real-world interpretability settings?

---

### Note · Authors · 2025-11-25

I have read and agree with the venue's withdrawal policy on behalf of myself and my co-authors.